# SELF-REFLECTIVE REINFORCEMENT LEARNING FOR DIFFUSION-BASED IMAGE REASONING GENERATION

## ABSTRACT

Recent years have witnessed the wonderful effects of diffusion models in generative task. However, existing image generation methods still suffer from the significant *"reasoning-oriented generative"* dilemma. Motivated by the success of Chain of Thought (CoT) and Reinforcement Learning (RL) in LLMs, we propose SRRL, a self-reflective RL algorithm for diffusion models to achieve reasoning generation of logical images by performing reflection and iteration across generation trajectories. The intermediate samples in the denoising process carry noise, making accurate reward evaluation difficult. To address this challenge, SRRL treats the entire denoising trajectory as a CoT step with multi-round reflective denoising process and introduces condition guided forward process, which allows for reflective iteration between CoT steps. Through SRRL-based iterative diffusion training, we introduce image reasoning through CoT into generation tasks adhering to physical laws and unconventional physical phenomena for the first time. Notably, experimental results of case study exhibit that the superior performance of our SRRL algorithm even compared with advanced T2I models.

## 1 INTRODUCTION

Recent years have witnessed the remarkable success of text-to-image (T2I) models (Ho et al., 2020; Ramesh et al., 2021; 2022). As the pioneering model among many T2I models, diffusion models have demonstrated powerful abilities in generating realistic images (Saharia et al., 2022; Ramesh et al., 2022; Podell et al., 2023; Ma et al.; Zhang et al., 2023; Ruiz et al., 2023; Ma et al., 2024). Existing works introduce ControlNet (Zhang et al., 2023) and T2I-Adapter (Mou et al., 2024) to enhance the controllability of image generation. However, these models still lack the ability of reflective reasoning, resulting in issues that images do not adhere to physical laws, where images may be visually stunning but logically inconsistent (Jiang et al., 2024; Huang et al., 2023).

Reinforcement learning (RL) based training methods (Black et al., 2023; Wallace et al., 2024; Fan et al., 2023; Hu et al., 2025; Majumder et al., 2024), including Direct Preference Optimization (DPO) and Proximal Policy Optimization (PPO), have recently been integrated into diffusion models to enhance specific capabilities, such as text-image alignment and human feedback alignment. DPO aligns diffusion models to human preferences by directly optimizing on comparison data, relying on high-quality user feedback, which leads to high collection costs. PPO optimizes the parameters by considering the step-by-step denoising process as a multi-step decision-making process (Black et al., 2023), which treats noisy samples at each timestep as states, denoising process at each timestep as actions, and evaluated score of the final images as rewards. However, PPO optimizes the entire trajectory according to the final images by outcome reward models (ORMs), lacking the ability for reflective reasoning, which results in insufficient capabilities of complex logical image generation.

Reflective reasoning through Chain-of-thought (CoT) (Wei et al., 2022) has been widely explored in LLMs by allowing models to decompose complex problems into several intermediate reasoning steps (Guo et al., 2025a; Jaech et al., 2024; Hui et al., 2024; Lightman et al., 2023). Despite CoT being widely used in LLMs to increase the ability of solving complex NLP problems, there is relatively less work (Guo et al., 2025b; Jiang et al., 2025a) on enhancing reasoning capabilities in the field of image generation. Very recently, some works (Guo et al., 2025b; Jiang et al., 2025a) explore CoT in auto-regressive image generation architecture. However, there remains a significant challenge, which is exploring introducing CoT into diffusion models to enhance image reasoning capabilities. The

Figure 1: Illustration of self-reflective reasoning step. Through self-reflective processes of repeated denoising and re-noising, diffusion models achieve image reasoning generation adhered to physical laws and counterintuitive physical phenomena.

step-by-step diffusion denoising process produces noisy intermediate samples that are difficult to evaluate, thereby hindering the implementation of CoT reasoning during the denoising process.

In this paper, we present a novel self-reflective RL algorithm **SRRL** of diffusion models, introducing CoT into diffusion models to provide self-reflective capabilities by RL training to achieve image reasoning generation. Specifically, SRRL incorporates multi-round reflective denoising process and condition guided forward process, treating the entire diffusion denoising trajectory as a step and constructing CoT between different trajectories instead of in a single denoising trajectory, which avoids the challenges of predicting rewards of noisy samples. Illustration of self-reflective reasoning step is shown in Fig. 1. With self-reflective capabilities, SRRL achieves image reasoning generation—for instance, ensuring that generated images adhere to physical laws, such as depicting plants growing taller with sunlight compared to those without in Fig. 3. Experimental results demonstrate that diffusion models trained by SRRL can generate images adhering to physical laws and counterintuitive scenarios. More impressively, images adhering to physical laws and counterintuitive physical phenomena generated through self-reflective reasoning of SRRL rival or surpass those generated by advanced T2I models recently.

Our contributions can be summarized as:

- We introduce a self-reflective RL algorithm SRRL, enabling diffusion models with the ability for self-reflective thinking and imagination.

- We explore introducing CoT into the generation process of diffusion models, allowing process reward models (PRMs) to address the issue of diffusion models being unable to self-reflect based on noisy intermediate results.

- Experimental results indicate that SRRL achieves image reasoning generation adhering to both physical laws and counterintuitive physical phenomena. Specifically, experimental samples of SRRL exhibit superior quality even compared to advanced T2I models recently.

## 2 RELATED WORK

### 2.1 TEXT-TO-IMAGE DIFFUSION MODELS

Diffusion models are widely used in text-to-image (T2I) tasks due to their exceptional performance in generating high-quality images (Song et al., 2020b; Dhariwal & Nichol, 2021; Nichol & Dhariwal, 2021; Song & Ermon, 2020). Diffusion models generate images by denoising noisy images under the guidance of the text conditions. Many works, such as Stable Diffusion (Rombach et al., 2022), Imagen (Saharia et al., 2022), DALL-E (Ramesh et al., 2021), GPT-4o (Hurst et al., 2024), demonstrate the ability of diffusion models in T2I tasks. The alignment between text and images has become an important metric for improving the effectiveness of the model. Classifier-free guidance (CFG) (Ho & Salimans, 2022) is introduced into diffusion models to enhance text conditions and text-image alignment. Some works (Chung et al., 2024; Bradley & Nakkiran, 2024; Li et al., 2024) improve the generation quality and text-image alignment by optimizing CFG. Zigzag diffusion sampling (Bai et al., 2024) incorporates a self-reflection mechanism leveraging CFG to accumulate semantic information during inference process. However, they do not consider allowing models to learn reasoning, which leads to their inability to generate logical images adhering to physical laws.

## 2.2 REINFORCEMENT LEARNING OF DIFFUSION MODELS

Reinforcement Learning from Human Feedback (RLHF) (Ouyang et al., 2022) is employed for better alignment of diffusion models to human preferences. Some reward models (Hessel et al., 2021; Xu et al., 2023; Lin et al., 2024) are trained to enhance aesthetic quality, text-image alignment, and so on, to align with human preferences. Diffusion denoising process can be seen as a sequential decision-making problem (Black et al., 2023), allowing the application of RL algorithms (Black et al., 2023; Fan et al., 2023; Hu et al., 2025; Ren et al., 2024). DDPO (Black et al., 2023) is a policy gradient algorithm treating diffusion denoising process as Markov decision process and using proximal policy optimization (PPO) (Schulman et al., 2017) updates. However, these algorithms use outcome reward models (ORMs) due to the challenge of evaluating intermediate noisy images and cannot self-reflective reasoning based on a single denoising process.

## 2.3 REFLECTIVE REASONING THROUGH CHAIN-OF-THOUGHT

Large language models (LLMs) and multi-modal large language models (MLLMs) are discovered to simulate human thought process by reflective reasoning based on their understanding and generation skills (Meng et al., 2025; Yang et al., 2025; Jiang et al., 2025b). Recent works (Jaech et al., 2024; Guo et al., 2025a; Wei et al., 2022) incorporate Chain-of-Thought (CoT) to achieve superior performance in text generation tasks, such as mathematics (Zhang et al., 2024; Lu et al., 2023), coding (Jain et al., 2024), and image understanding (Huang et al., 2025) problems. On the contrary, the exploration of CoT in image generation has been more limited. Some works (Guo et al., 2025b; Jiang et al., 2025a) explore incorporating CoT in image generation tasks. However, it uses the auto-regressive architecture as the backbone, without exploring the potential of CoT in T2I diffusion models, which are more widely used in commercial applications.

## 3 METHOD

In this section, we first introduce the training of diffusion models using reinforcement learning (RL) algorithms and self-reflective RL algorithm of diffusion models SRRL in Sec. 3.1. Then we propose multi-round reflective denoising process in Sec. 3.2 and condition guided forward process in Sec. 3.3. These two processes together constitute SRRL algorithm, which is illustrated in Fig. 2.

### 3.1 PROBLEM FORMULATION

#### 3.1.1 REINFORCEMENT LEARNING TRAINING OF DIFFUSION MODELS

Text-to-image diffusion models generate images by progressively denoising noisy images. We follow the formulation of diffusion models in denoising diffusion probabilistic models (DDPMs) (Ho et al., 2020). Diffusion models are composed of two processes: forward process and denoising process.

**Forward Process**. Given a dataset with samples $x_0 \sim q_0(x_0|c)$ where $q_0$ is the data distribution and corresponding to text condition $c$, forward process is a Markov chain that gradually adds Gaussian noise into $x_0$ in T timesteps according to the variance schedule $\beta_t$:

$$q(x_t|x_{t-1}) = \mathcal{N}(x_t; \sqrt{1-\beta_t}x_{t-1}, \beta_t \mathbf{I}), \qquad q(x_{1:T}|x_0) = \prod_{t=1}^{T} q(x_t|x_{t-1}) \qquad (1)$$

Forward process constructs an approximate posterior distribution, and the goal of denoising process is to approximate it.

**Denoising Process** is a Markov chain, which can be seen as a Markov decision process (MDP).

$$p_\theta(x_{t-1}|x_t, c) = \mathcal{N}(x_{t-1}; \boldsymbol{\mu}_\theta(x_t, c, t), \boldsymbol{\Sigma}_t), \qquad p_\theta(x_{0:T}|c) = p(x_T) \prod_{t=1}^{T} p_\theta(x_{t-1}|x_t, c) \qquad (2)$$

where $\boldsymbol{\mu}_\theta(x_t, c, t)$ is predicted by a diffusion model $\theta$, and $\Sigma_t$ is variance related to timestep $t$. Given samples $x_0$ and text condition $c \sim p(c)$, text-to-image diffusion models generate images according to text condition $c$. Classifier-free guidance (CFG) (Ho & Salimans, 2022) enhances text conditions to improve image generation quality by subtracting the predicted unconditional noise from the conditional noise:

$$\tilde{\epsilon}_\theta(x_t, c, t, \lambda) = \epsilon_\theta(x_t, c, t) + \lambda(\epsilon_\theta(x_t, c, t) - \epsilon(x_t, \phi, t)) \qquad (3)$$

Here $\epsilon_\theta(x_t, c, t)$ is the conditional noise satisfying $\mu_\theta(x_t, t, c) = \frac{1}{\sqrt{\alpha_t}}(x_t - \frac{\beta_t}{\sqrt{1-\bar{\alpha}_t}}\epsilon_\theta(x_t, c, t))$, where $\alpha_t = 1 - \beta_t, \bar{\alpha}_t = \prod_{i=1}^{t} \alpha_i$ (Ho et al., 2020). $\phi$ refers to no condition during the denoising process.

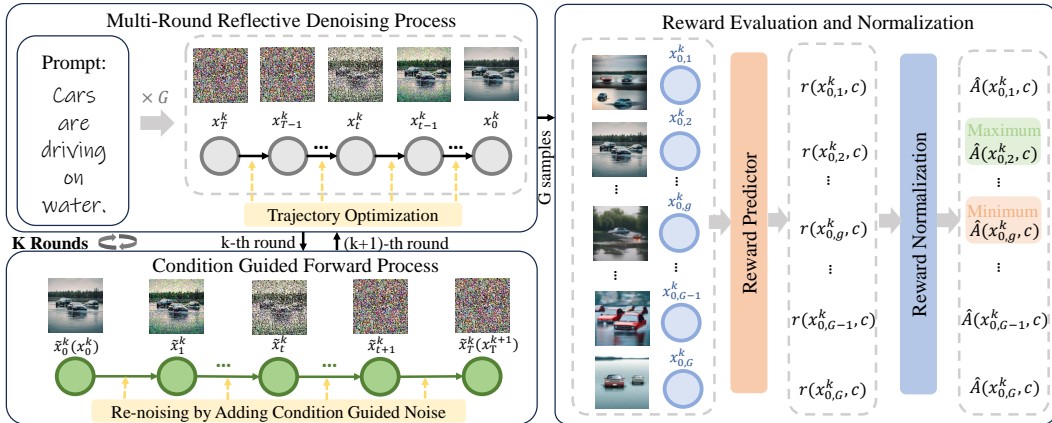

Figure 2: Overview of SRRL. SRRL includes two processes: multi-round reflective denoising process and condition guided forward process. These two processes are repeated for $K$ rounds.

The goal of DDPMs is approximating $q_0(x_0|c)$ with $p_\theta(x_0|c) = \int p_\theta(x_{0:T}|c)dx_{1:T}$. The denoising process can be seen as a multi-step MDP $\tau = (s_T, a_T, s_{T-1}, a_{T-1}, \cdots, s_0, a_0)$:

$$s_t = (c, t, x_t), \qquad a_t = x_{t-1}, \qquad \pi_\theta(a_t|s_t) = p_\theta(x_{t-1}|x_t, c), \qquad R(s_t, a_t) = \begin{cases} r(x_0, c), & \text{if } t = 0 \\ 0, & \text{otherwise} \end{cases}$$

where $s_t$ is the state at each timestep, $a_t$ is the action to denoise $x_t$ to $x_{t-1}$, $\pi_\theta$ defines the action selection strategy, and $R$ is the reward, which is given by models or human preferences. Therefore, the diffusion denoising process can be viewed as an RL task in which diffusion models act as agents to make decisions (denoising process). The goal of RL is to maximize the expected cumulative reward over the diffusion denoising trajectories sampled from the policy, which can be formulated as:

$$\mathcal{J}_{RL}(\theta) = \mathbb{E}_{c \sim p(c), x_0 \sim p_\theta(x_0|c)}[r(x_0, c)] \tag{4}$$

where $p(c)$ is the distribution of text descriptions of images.

### 3.1.2 SELF-REFLECTIVE REINFORCEMENT LEARNING

Existing reinforcement learning algorithms (Black et al., 2023; Fan et al., 2023; Hu et al., 2025) optimize only a single denoising trajectory and can only utilize outcome reward models (ORMs) without reflective reasoning capabilities. Different from them, SRRL aims to optimize the cumulative denoising trajectory, enabling it to utilize process reward models (PRMs) from intermediate results, which enables self-reflective reasoning process. The objective of SRRL is:

$$\mathcal{J}_{SRRL}(\theta) = \mathbb{E}_{c \sim p(c), x_0 \sim p_\theta(x_0|c), k \sim U(0,K)}[r(x_0^k, c)] \tag{5}$$

where $k$ refers to the $k$-th iteration of the reflection process, $x_0^k$ refers to the $k$-th intermediate sample for evaluation, $U$ refers to uniform distribution. SRRL includes multi-round reflective denoising process and condition guided forward process, which will be detailed in the following sections.

### 3.2 MULTI-ROUND REFLECTIVE DENOISING PROCESS

Diffusion models suffer from the issue that reward prediction is limited to final images, preventing the introduction of PRMs and resulting in a lack of reflection capability. To address the issue, SRRL incorporates multiple rounds of RL optimization in the denoising process, providing PRMs and aiming to endow the model with self-reflection capability. Specifically, after each round of the denoising process, SRRL evaluates intermediate images using reward models, which provide process rewards for the entire multi-round process. In the subsequent rounds, SRRL optimizes the trajectory based on the intermediate rewards from previous rounds.

SRRL leverages policy gradient estimation by computing likelihoods and gradients of likelihoods:

$$\nabla_\theta \mathcal{J}_{SRRL} = \mathbb{E}_{c \sim p(c), x_0 \sim p_\theta(x_0|c), k \sim U(0,K)}[\sum_{t=0}^{T_k} \nabla_\theta \log p_\theta(x_{t-1}^k|x_t^k, c) r(x_0^k, c)] \tag{6}$$

Evaluation of the above requires sampling from the multi-round denoising process, which can be seen as a long MDP $\tau_{SRRL} = (s_T^0, a_T^0, \cdots, s_0^0, a_0^0, \cdots, s_T^k, a_T^k, \cdots, s_0^k, a_0^k, \cdots, s_0^K, a_0^K)$. The reward includes process rewards of intermediate samples: $R(s_t^k, a_t^k) = \begin{cases} r(x_0^k, c), & \text{if } t = 0 \\ 0, & \text{otherwise} \end{cases}$.

We apply Proximal Policy Optimization (PPO) (Schulman et al., 2017) algorithm, including importance sampling and clipping. Besides, we use reward normalization and remove the value function, similar to Group Relative Policy Optimization (Shao et al., 2024) algorithm, and contrastive sampling (Mikolov et al., 2013) is introduced. The PPO update objective is:

$$\nabla_\theta \mathbb{E}_{c \sim p(c), k \sim U(0,K)} \frac{1}{G_c} \sum_{i=1}^{G_c} \left( \sum_{t=1}^{T} [\min(\frac{p_\theta(x_{t-1}^k | x_t^k, c)}{p_{old}(x_{t-1}^k | x_t^k, c)} \hat{A}_i^k, \text{clip}(\frac{p_\theta(x_{t-1}^k | x_t^k, c)}{p_{old}(x_{t-1}^k | x_t^k, c)}, 1 - \epsilon, 1 + \epsilon) \hat{A}_i^k)) \right)$$

where $G_c$ is the number of remaining samples after contrastive sampling (selecting the maximum and minimum reward values). $\hat{A}_i^k$ is calculated through reward normalization: $\hat{A}_i^k = \frac{r(x_{0,i}^k, c) - \text{mean}(\{r(x_{0,1}^k, c), \cdots, r(x_{0,G}^k, c)\})}{\text{std}(\{r(x_{0,1}^k, c), \cdots r(x_{0,G}^k, c)\})}$, where $G$ is the number of samples before contrastive sampling and $k$ is the $k$-th reflection round.

### 3.3 CONDITION GUIDED FORWARD PROCESS

By optimizing multi-round denoising process, SRRL gains self-reflection ability through PRMs. However, a problem is how to connect the multiple rounds of denoising processes, allowing reflective iteration between image CoT steps. To achieve multi-round self-reflection between different denoising trajectories, SRRL proposes condition guided forward process, which adds conditional noise to intermediate samples at the end of each denoising round to obtain noisy samples for the next round of reflective denoising process.

Given the intermediate sample $x_0^k$, the condition guided forward process aims to add noise to obtain the noisy sample $x_T^{k+1}$ of the next round, which can be formulated as:

$$x_T^{k+1} = \prod_{t=1}^{T} \chi(\tilde{x}_t^k | \tilde{x}_{t-1}^k, c), \quad k = 0, 1, \cdots, K$$

$$\chi(\tilde{x}_t^k | \tilde{x}_{t-1}^k, c) = \sqrt{\frac{\alpha_t}{\bar{\alpha}_{t-1}}} \tilde{x}_{t-1}^k + (\frac{1 - \alpha_t}{\sqrt{1 - \bar{\alpha}_t}} - \sqrt{\frac{\alpha_t(1 - \bar{\alpha}_{t-1})}{\bar{\alpha}_{t-1}}}) \tilde{\epsilon}_\theta(\tilde{x}_{t-1}^k, c, t, \lambda)$$

(7)

where $x_T^{k+1} = \tilde{x}_T^k$ and $x_0^k = \tilde{x}_0^k$. SRRL sets CFG guidance scale $\lambda$ in forward process smaller than that in denoising process, e.g., 1.0 and 4.5. By creating a guidance gap between forward process and denoising process, SRRL injects text condition during forward process, leading to progressively better results with more reflection rounds. SRRL uses DDIM (Song et al., 2020a) inversion scheduler, which is a deterministic sampling method to precisely inject text conditions.

In summary, SRRL optimizes the denoising trajectory over multiple rounds and introduces intermediate sample reward evaluations, which address the issue that reward prediction is limited to final images. Besides, by introducing condition guided forward process, SRRL establishes inter-trajectory CoT connections, enabling iterative reflection and knowledge transfer across sequential steps. Multiple rounds of the denoising and forward process provide self-reflection ability, facilitating image reasoning generation in diffusion models. The pseudo-codes of training and inference process of SRRL are shown in Algorithm 1 and Algorithm 2.

## 4 EXPERIMENTS

In this section, we evaluate SRRL's effectiveness in image reasoning generation tasks. We aim to answer the following questions: i) Is it possible to leverage a self-reflective reinforcement learning algorithm to achieve image reasoning generation adhered to physical laws and unconventional physical phenomena? ii) How do generated images include reasoning and thought processes?

### 4.1 EXPERIMENTAL SETUP

**T2I diffusion models**. We use Stable Diffusion (SD) v1.4 (Rombach et al., 2022), SD XL (Podell et al., 2023) and SD 3 Medium (Esser et al., 2024) as the backbone diffusion models, which are open-source and widely used for T2I tasks. We perform LoRA (Hu et al., 2022) fine-tuning on U-Net in SD, which is a method that saves GPU memory and accelerates training efficiency. During multi-round reflective denoising process, SRRL uses DDIM (Song et al., 2020a) scheduler. During condition guided forward process, SRRL uses DDIM inversion scheduler. The number of sampling steps is set to 20. The implementation details are shown in Appendix 7.

Round ↑

Student testing friction with objects on ramps.

Reflection Process: Initially, a plank is horizontal with no object on it. Then, objects gradually appear on an inclined plank, enabling a student to explore friction. (related to physical laws)

Scientist comparing plant growth with and without sunlight.

Reflection Process: Initially, plants get equal light. Over time, sunlight exposure varies, causing height differences—taller with sunlight, shorter without—consistent with biological principles.

Person reflecting light using mirrors onto targets.

Reflection Process: Initially, mirrors are misshapen or do not reflect light. Later, they reflect light evenly. Eventually, the mirrors reflect light with varying intensity, consistent with optical laws.

Figure 3: Reasoning generation of images related to physical laws.

**Reward models and metrics**. We use CLIP Score (Hessel et al., 2021), ImageReward (Xu et al., 2023), and VQAScore (Lin et al., 2024) to evaluate the text-image alignment and image reasoning abilities of models. CLIP Score measures the similarity between text and image embeddings via CLIP model (Radford et al., 2021), trained with contrastive learning for cross-modal alignment. Image reward (Xu et al., 2023) evaluates the general-purpose text-to-image human preference by training on total 137k pairs text-images with expert comparisons. VQAScore (Lin et al., 2024) employs a visual-question-answering model to compute an alignment score. This is achieved by measuring the probability of the model responding 'Yes' to the question: 'Does this figure depict {text}?'. VQAScore is better in evaluating image reasoning ability due to its judgment ability.

**Prompt type**. We evaluate the effectiveness of our algorithm on three types of prompts. i) Following previous works (Black et al., 2023; Hu et al., 2025), we use the prompt template "a(n) [animal] [activity]", which evaluates text-image alignment. There are 45 kinds of animals and three activities: "riding a bike", "playing chess", and "washing dishes". Animals and activities are randomly matched. ii) Physical phenomenon-related prompts. These prompts include knowledge about physical laws. Details are shown in Appendix 11. iii) Unconventional physical phenomena prompts. These prompts contradict common phenomena to evaluate models' imagination capabilities. Details are shown in Appendix 11. It is worth noting that image reasoning capability and image-text alignment are not equivalent, and we discuss it in Sec. 5.

### 4.2 PHYSICAL LAW RELATED IMAGE GENERATION

We train models with the SRRL algorithm using prompts related to physical phenomena. Fig. 3 shows some qualitative results. The first prompt is "Student testing friction with objects on ramps". Initially, generated images lack inclined planks, and objects on the plank are unclear. With iterative self-reflection training, the final image includes an inclined plank with clear objects on it, depicting a student testing friction. The second prompt is "Scientist comparing plant growth with and without sunlight", which contains biological principles: plants receiving adequate sunlight grow better than those that do not. At first, two plants are similar. Gradually, the model learns to differentiate the intensity of light exposure on plants. Eventually, the model realizes that plants exposed to more light grow better. The third prompt is "Person reflecting light using mirrors onto targets". It indicates that the light on different mirrors is different. Initially, the mirrors are misshapen or do not reflect any light. Later, they reflect light evenly. Eventually, the mirrors reflect light with varying intensity,

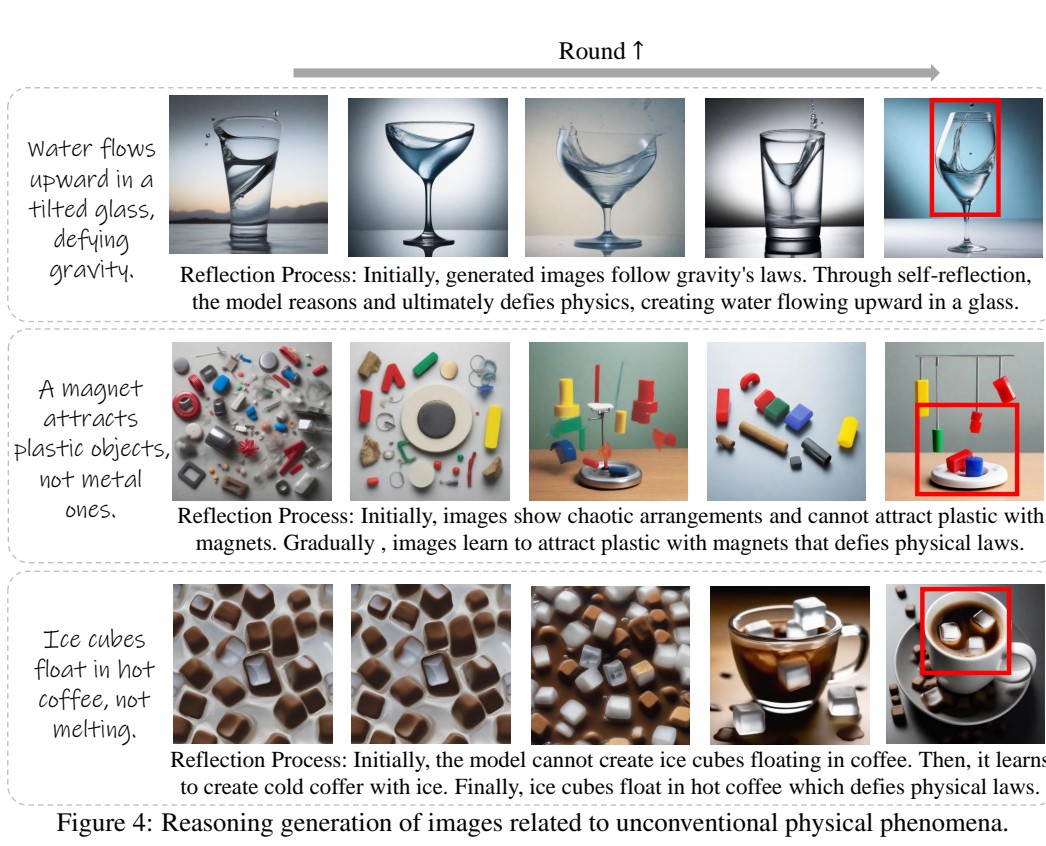

Figure 4: Reasoning generation of images related to unconventional physical phenomena.

consistent with physical laws. The above results indicate that SRRL, through self-reflection, can gradually learn to reason and generate images following physical laws.

### 4.3 UNCONVENTIONAL IMAGE GENERATION

We train models with SRRL algorithm using prompts for unconventional physical phenomena, which are counterintuitive and contradict usual physical phenomena. Some qualitative results are shown in Fig. 4. The first prompt is "Water flows upward in a tilted glass, defying gravity". At first, generated images obey physical laws of gravity. As the self-reflection process continues, the model engages in reasoning and eventually overcomes physical laws, generating an image of water flowing upwards in a glass. The second prompt is "A magnet attracts plastic objects, not metal ones". Initially, the objects in the images are chaotic, indicating that the model does not know how to use a magnet to attract plastic objects. Through self-reflection process, the model learns to attract plastic objects with a magnet, even though this defies physical laws. The third prompt is "Ice cubes float in hot coffee, not melting". The model initially cannot generate ice cubes floating in hot coffee. As the reasoning process progresses, the model learns this concept. From the bubbling coffee in the image, it can be inferred that the coffee is hot, while the ice cubes in the coffee have not melted. From the results above, it can be observed that initially, the model either adheres to physical laws or lacks relevant knowledge. As self-reflection activates its reasoning abilities, the model is able to generate images that defy common sense or physical laws.

### 4.4 VISUALIZATION OF IMAGE REASONING PROCESS

To visualize the reasoning process of SRRL, we incorporate two prompts: "Draw a balance. The object on the left side is lighter than that on the right side, but the balance leans to the left" and "Cars are driving on water", which contains contradictory knowledge. Reasoning process visualizations of the prompt related to the balance is shown in Fig. 5. We train SD XL model (Podell et al., 2023) with SRRL algorithm on one prompt each time and we use ImageReward (Xu et al., 2023) as the reward model. For results of the first prompt in Fig. 5, initially, the model generates images of a balance either tilted left with no objects or tilted right with lighter objects on the left and heavier ones on the right, both aligning with common sense or physical laws. Eventually, the model produces an image with contradictory elements: the balance is tilted left despite having no objects on the left and a small ball on the right. More reasoning process visualization and analysis are shown in Appendix 10. This

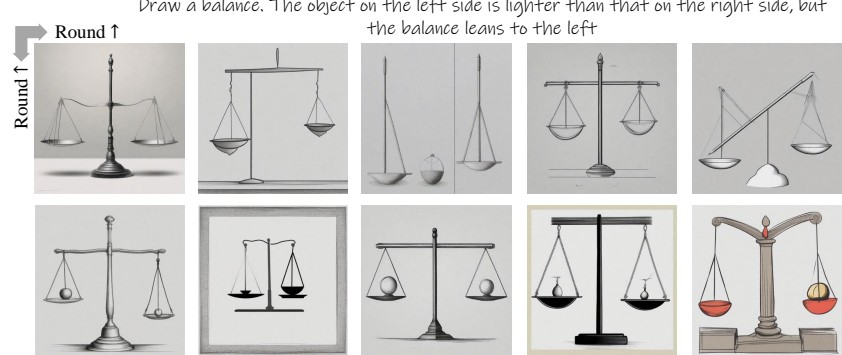

Figure 5: Reasoning generation process of the prompt related to a balance. Initially, the model generates an image of a balance tilted left without objects or tilted right with lighter objects on the left and heavier ones on the right, both following physical laws. Eventually, it learns to create images defying logic: a balance tilts left with no objects on the left and a small ball on the right.

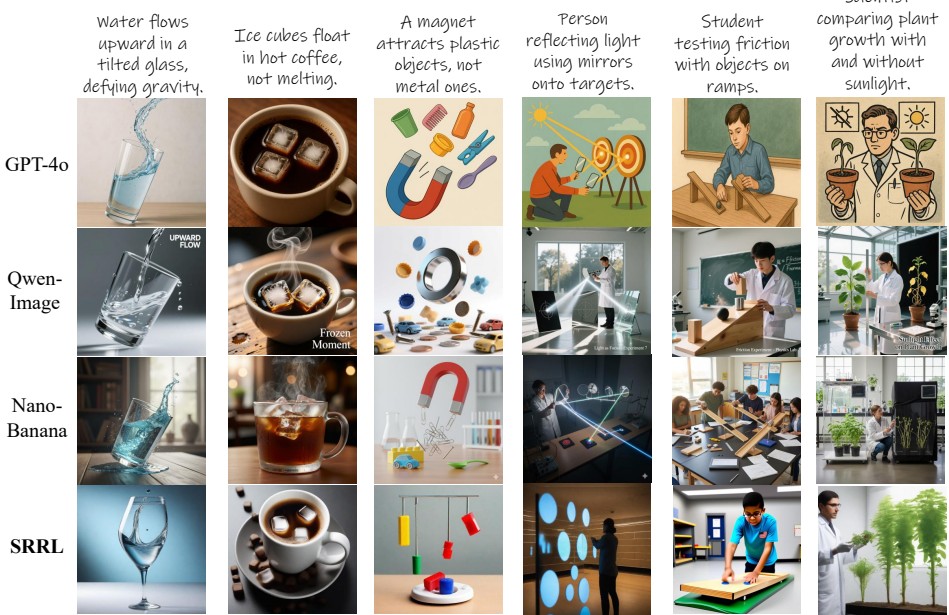

Figure 6: Example of generated images of physical phenomenon related prompts and unconventional physical phenomena prompts by advanced T2I models and SRRL.

suggests that by introducing self-reflection mechanism, SRRL performs reasoning and can generate images adhering to contradictory common sense, showing the model's imagination ability.

### 4.5 COMPARATIVE ANALYSIS AND ABLATION STUDY

**Comparison with Advanced T2I Models**. We compare samples of SRRL with those of GPT-4o (Hurst et al., 2024), Qwen-Image (Wu et al., 2025), and Nano-Banana (Fortin et al., 2025), which are most advanced T2I models recently in Fig. 6. SRRL generates similar or higher quality images compared to other advanced models, showing reasoning capabilities akin to other models.

**Evaluation of SRRL on Physical-Law and Unconventional Prompts**. We evaluate the SRRL algorithm using prompts related to physical laws and unconventional phenomena. The results are presented in Table 1. In comparison to vanilla Stable Diffusion (SD) and DDPO (Black et al., 2023), SRRL consistently achieves superior performance across various reward models.

**Comparison on "Animal–Activity" Prompts with Prior Works**. We also train SD v1.4 model using SRRL algorithm on the prompt template "a(n) [animal] [activity]" to compare with previous works (Black et al., 2023; Hu et al., 2025). Quantitative results of prompt template "a(n) [animal] [activity]" are shown in Tab. 2. Compared with baseline, SRRL performs better in ImageReward and VQAScore. Appendix 9 shows some qualitative results of the prompt template. Compared to

Table 1: Quantitative results on physical laws and unconventional phenomena.

| Image Type | Physical Laws | | | Unconventional Phenomena | | |
|---|---|---|---|---|---|---|
| Reward Models | CLIPScore | ImageReward | VQAScore | CLIPScore | ImageReward | VQAScore |
| SD v1.4 (w/o RL) | 0.302 | 0.281 | 0.559 | 0.296 | 0.275 | 0.520 |
| SD v1.4+DDPO | 0.321 | 0.297 | 0.628 | 0.321 | 0.318 | 0.562 |
| SD v1.4+SRRL | **0.385 (+27.5%)** | **0.369 (+31.3%)** | **0.623 (+11.4%)** | **0.329 (+11.1%)** | **0.327 (+18.9%)** | **0.584 (+12.3%)** |
| SD XL (w/o RL) | 0.326 | 0.293 | 0.591 | 0.301 | 0.287 | 0.544 |
| SD XL+DDPO | 0.339 | 0.305 | 0.637 | 0.330 | 0.323 | 0.589 |
| SD XL+SRRL | **0.349 (+7.1%)** | **0.324 (+10.6%)** | **0.655 (+7.8%)** | **0.335 (+11.3%)** | **0.331 (+15.3%)** | **0.632 (+16.2%)** |
| SD 3 (w/o RL) | 0.357 | 0.311 | 0.613 | 0.339 | 0.315 | 0.610 |
| SD 3+DDPO | 0.372 | 0.339 | 0.648 | 0.345 | 0.329 | 0.626 |
| SD 3+SRRL | **0.413 (+15.7%)** | **0.368 (+18.3%)** | **0.689 (+12.4%)** | **0.397 (+17.1%)** | **0.371 (+17.7%)** | **0.674 (+10.5%)** |

Table 2: Quantitative results of the prompt template "a(n) [animal] [activity]".

| Methods | SD v1.4 | DDPO (Black et al., 2023) | B2-DiffuRL (Hu et al., 2025) | SRRL (Ours) |
|---|---|---|---|---|
| CLIP Score ↑ | 0.3624 | **0.3683** | 0.3674 | 0.3662 |
| ImageReward ↑ | 0.2823 | 0.3534 | 0.3682 | **0.3807** |
| VQAScore ↑ | 0.6045 | 0.6145 | 0.6174 | **0.6338** |

baselines, SRRL generates images that are more prompt-aligned and of higher quality, indicating that self-reflection mechanism also enhances models' text-to-image alignment ability.

**Reflection Rounds and Refinement in Image Generation**. We assess the performance of SRRL across prompts concerning physical phenomena and unconventional physical scenarios, with respect to the increasing number of reflection rounds, and results are shown in Fig. 7. The quality of generated images improves as the round increases with self-reflection process. We also notice the phenomenon of reflection refinement, which involves adjusting the generated images through image reconstruction.

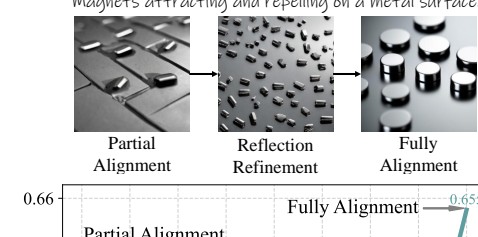

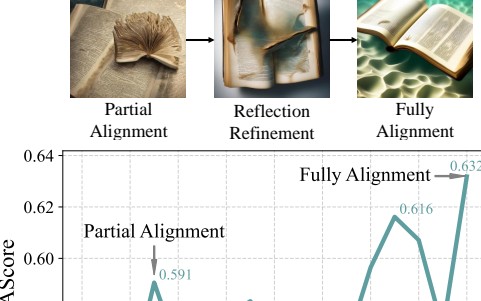

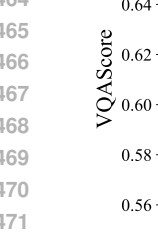

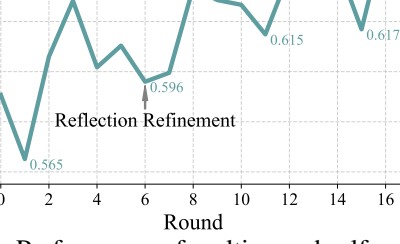

Figure 7: Performance of multi-round self-reflection of SRRL. The left is results of physical phenomenon related prompts, and the right is those of unconventional physical phenomena prompts. The figure above shows some cases of reflection process. All experiments are done based on SD XL.

## 5 DISCUSSION AND CONCLUSION

Chain-of-thought (CoT) has recently been applied to LLMs to enhance self-reflection in complex tasks. Extending CoT to image generation raises the question of which challenges it should address. We focus on generating images consistent with physical laws, as this setting highlights models' reasoning and imaginative capacity. Unlike text–image alignment, adherence to physical laws is difficult because such constraints are often implicit in textual prompts.

We introduce SRRL, a self-reflective reinforcement learning algorithm for diffusion models that incorporates image CoT and multi-round reflective denoising with condition-guided forward processes. Experiments demonstrate that SRRL produces images consistent with physical laws and unconventional scenarios, thereby advancing image reasoning.

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

# 6 DERIVATIONS

## 6.1 EQUATION 6

$$
\begin{aligned}
\nabla_\theta[-\mathcal{J}_{SRRL}(\theta)] &= \nabla_\theta \mathbb{E}_{p(c)} \mathbb{E}_{p_\theta(x_0|c)} \mathbb{E}_{k \sim U(0,K)}[-r(x_0^k, c)] \\
&= -\mathbb{E}_{p(c)} \mathbb{E}_{k \sim U(0,K)}[\nabla_\theta \int r(x_0^k, c) p_\theta(x_0^k|c) dx_0^k] \\
&= -\mathbb{E}_{p(c)} \mathbb{E}_{k \sim U(0,K)}[\nabla_\theta \int r(x_0^k, c)(\int p_\theta(x_{0:T}^k|c) dx_{1:T}^k) dx_0^k] \\
&= -\mathbb{E}_{p(c)} \mathbb{E}_{k \sim U(0,K)}[\int \nabla_\theta \log p_\theta(x_{0:T}^k|c) r(x_0^k, c) p_\theta(x_{0:T}^k|c) dx_{0:T}^k] \\
&= -\mathbb{E}_{p(c)} \mathbb{E}_{k \sim U(0,K)}[\int \nabla_\theta \log \left( p_T(x_T^k|c) \prod_{t=1}^T p_\theta(x_{t-1}^k|x_t^k, c) \right) r(x_0^k, c) p_\theta(x_{0:T}^k|c) dx_{0:T}^k] \\
&= -\mathbb{E}_{p(c)} \mathbb{E}_{p_\theta(x_{0:T}^k|c)} \mathbb{E}_{k \sim U(0,K)}[\sum_{t=1}^T \nabla_\theta \log p_\theta(x_{t-1}^k|x_t^k, c) r(x_0^k, c)]
\end{aligned}
$$

Here the proof uses the continuous assumptions of $p_\theta(x_{0:T}^k|c) r(x_0^k, c)$.

## 6.2 EQUATION 7

Following DDPM (Ho et al., 2020), the denoising process is formulated as:

$$
x_{t-1}^k = \frac{\sqrt{\bar{\alpha}_{t-1}}}{\sqrt{\alpha_t}}(x_t^k - \frac{1-\alpha_t}{\sqrt{1-\bar{\alpha}_t}}\epsilon_\theta(x_t^k, c, t, \lambda)) + \sqrt{1-\bar{\alpha}_{t-1}}\epsilon_\theta(x_t^k, c, t, \lambda) \tag{8}
$$

Then, solve for $x_t$ based on $x_{t-1}$,

$$
\tilde{x}_t^k = \sqrt{\frac{\alpha_t}{\bar{\alpha}_{t-1}}}\tilde{x}_{t-1}^k + (\frac{1-\alpha_t}{\sqrt{1-\bar{\alpha}_t}} - \sqrt{\frac{\alpha_t(1-\bar{\alpha}_{t-1})}{\bar{\alpha}_{t-1}}})\tilde{\epsilon}_\theta(\tilde{x}_{t-1}^k, c, t, \lambda) \tag{9}
$$

Here we leverage the assumption that $\tilde{\epsilon}_\theta(\tilde{x}_{t-1}^k, c, t, \lambda) \approx \tilde{\epsilon}_\theta(\tilde{x}_t^k, c, t, \lambda)$.

We can inject condition if there is a guidance gap between forward process and denoising process.

For convenience, we set:

$$
\gamma_t = \sqrt{\frac{\alpha_t}{\bar{\alpha}_{t-1}}}, \qquad \eta_t = (\frac{1-\alpha_t}{\sqrt{1-\bar{\alpha}_t}} - \sqrt{\frac{\alpha_t(1-\bar{\alpha}_{t-1})}{\bar{\alpha}_{t-1}}}) \tag{10}
$$

Then,

$$
\begin{aligned}
\tilde{x}_T^k &= \gamma_T \tilde{x}_{T-1}^k + \eta_T \tilde{\epsilon}_\theta(\tilde{x}_{T-1}^k, c, t, \lambda) \\
&= \gamma_T \gamma_{T-1} \tilde{x}_{T-2}^k + \gamma_T \eta_{T-1} \tilde{\epsilon}_\theta(\tilde{x}_{T-2}^k, c, t, \lambda) + \eta_T \tilde{\epsilon}_\theta(\tilde{x}_{T-1}^k, c, t, \lambda) \\
&= \cdots \\
&= \prod_{i=0}^T \gamma_i \tilde{x}_0 + \sum_{t=1}^T \eta_t \prod_{l=t+1}^T \gamma_l \epsilon_\theta(\tilde{x}_{t-1}^k, c, t, \lambda)
\end{aligned} \tag{11}
$$

Similarly, we can get:

$$
x_T^k = \prod_{i=0}^T \gamma_i x_0 + \sum_{t=1}^T \eta_t \prod_{k=t+1}^T \eta_k \epsilon_\theta(x_{t-1}^k, c, t, \lambda_{\text{Forward}}) \tag{12}
$$

The guidance gap can be formulated as:

$$
\begin{aligned}
\delta_k &= (x_T^k - \tilde{x}_T^k)^2 \\
&= [\prod_{i=0}^{T} \gamma_i (x_0 - \tilde{x}_0) + \sum_{t=1}^{T} \eta_t \prod_{l=t+1}^{T} \gamma_l (\epsilon_\theta(x_{t-1}^k, c, t, \lambda_{\text{Forward}}) - \epsilon_\theta(\tilde{x}_{t-1}^k, c, t, \lambda))]^2 \\
&= (\sum_{t=1}^{T} F(\eta_t, \gamma_t)(\epsilon_\theta(x_{t-1}^k, c, t, \lambda_{\text{Forward}}) - \epsilon_\theta(\tilde{x}_{t-1}^k, c, t, \lambda)))^2 \\
&= (\sum_{t=1}^{T} F(\eta_t, \gamma_t)(\lambda_{\text{Forward}} - \lambda)\epsilon_\theta(x_{t-1}^k, c, t))^2
\end{aligned}
\tag{13}
$$

By setting a guidance scale gap between $\lambda$ and $\lambda_{\text{Forward}}$, we can inject text condition during condition injection reflection forward process. Through multiple rounds of self reflection, the effect of condition injection is enhanced.

## 7 IMPLEMENTATION DETAILS

When fine-tuning Stable Diffusion model (Rombach et al., 2022; Podell et al., 2023) using LoRA according to SRRL algorithm, the configs are:

| Config | Value |
|---|---|
| LoRA rank | 4 |
| LoRA alpha | 4 |
| lr | 1e-4 |
| optimizer | Adam (Kingma & Ba, 2014) |
| weight decay of optimizer | 1e-4 |
| $\beta_1, \beta_2$ | (0.9,0.999) |
| number of samples per batch $G$ | 32 |
| self-reflection total rounds $K$ | 10 |
| denoising timestep $T$ | 20 |
| reward function $r$ | CLIP Score, ImageReward, VQAScore |
| training epoch number $E$ | 2 |
| forward guidance scale | 0.5 |
| denoising guidance scale | 3.0 |
| inference guidance scale | 7.5 |

# 8 PSEUDO-CODE OF SRRL

---

**Algorithm 1:** SRRL Training Process

---

**Input:** Pretrained diffusion model $p_\theta$, denoising timestep $T$, self-reflection total rounds $K$, number of samples per batch $G$, prompts list C, reward function $r$, training epoch number $E$.

1:  $k = 0$
2:  **repeat**
3:      $e = 0$
4:      **repeat**
5:          SampleList=[]
6:          $n = 0$
7:          **repeat**
8:              Random choose prompt c from C,
9:              Random sample Gaussian noise $x_T$ in $\mathcal{N}(0, 1)$,
10:             $i = 0$
11:             **repeat**
12:                 Denoise $x_T^i$ to $x_0^i$ with $p_\theta$,
13:                 Noise injection $x_0^i$ to $x_T^{i+1}$ with $p_\theta$,
14:                 $i = i + 1$
15:             **until** $i = k$
16:             $x_{T:0}^k = $ DDIM-Scheduler$_\theta(x_T^k \to x_0^k)$,
17:             SampleList.append([$x_{T:0}^k$,c]),
18:             $n = n + 1$
19:         **until** $n = G$
20:         Evaluate $r(x_{0,i=1:G}^k, c)$,
21:         score$_{i=1}^G$ = Reward Normalization($r(x_{0,i=1:G}^k, c)$,
22:         score$_{\max}, i_{\max}$ = maximum(score$_{i=1}^G$), index(score$_{\max}$),
23:         score$_{\min}, i_{\min}$ = minimum(score$_{i=1}^G$), index(score$_{\min}$),
24:         update $\theta$ according to [$x_{T:0,i_{\max}}^k$, score$_{\max}$, $x_{T:0,i_{\min}}^k$, score$_{\min}$, c].
25:         $e = e + 1$
26:     **until** $e = E$
27:     $k = k + 1$
28: **until** $k = K$

**Output:** Fine-tuned Model $p_{\theta'}$

---

---

**Algorithm 2:** SRRL Inference Process

---

**Input:** Fine-tuned diffusion model $p_\theta$, self-reflection rounds $k$

1: Random sample Gaussian noise $x_T^0$ in $\mathcal{N}(0, 1)$,
2: $i = 0$
3: **repeat**
4:     Denoise $x_T^i$ to $x_0^i$ with $p_\theta$,
5:     Noise injection $x_0^i$ to $x_T^{i+1}$ with $p_\theta$,
6:     $i = i + 1$
7: **until** $i = k$
8: Denoise $x_T^k$ to $x_0^k$

**Output:** Self-reflective images $x_0^k$

---

# 9 QUALITATIVE COMPARISON ON "ANIMAL–ACTIVITY" PROMPTS

Qualitative results on "Animal–Activity" Prompts of SRRL, DDPO (Black et al., 2023), B2-DiffuRL (Hu et al., 2025), and vanilla SD are shown in Fig. 8.

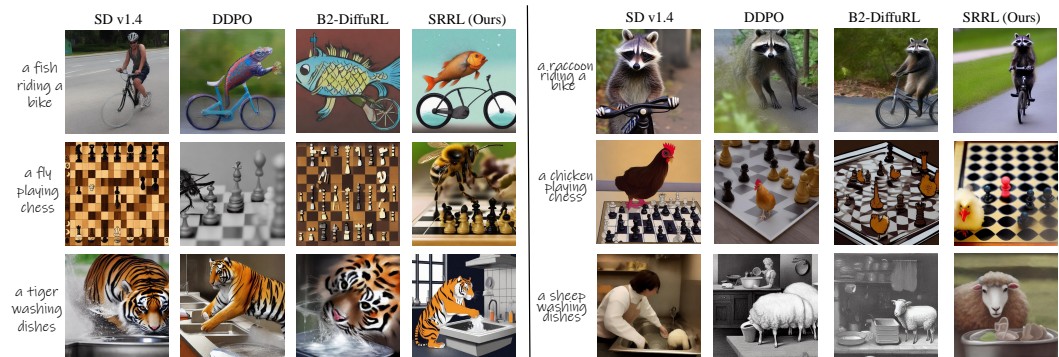

Figure 8: Examples of prompt template "a(n) [animal] [activity]" by baselines (Rombach et al., 2022; Black et al., 2023; Hu et al., 2025) and SRRL.

## 10  ADDITIONAL VISUALIZATION OF IMAGE REASONING PROCESS

Additional visualization of image reasoning process is shown in Fig. 9. For results of the prompt in Fig. 9, the general common sense is that cars drive on land, but the prompt requires generating an image of a car driving on water. In the initially generated images, the car appears to fly out of the water. In the final generated images, the car drives in the center of the lake rather than floats.

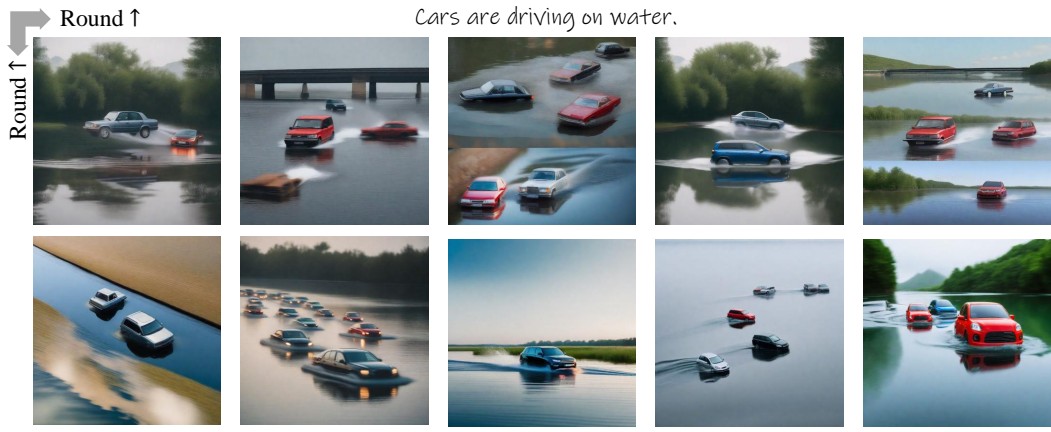

Figure 9: Reasoning generation process of prompt related to cars. Common sense dictates that cars drive on land, but the prompt asks for an image of a car on water. Initially, cars seems to fly out of the water, but in the final images, they drive across the center of the lake rather than float.

## 11  PROMPT DETAILS

### 11.1  PROMPTS TEMPLATE "A(N) [ANIMAL] [ACTIVITY]"

Prompts of the template "a(n) [animal] [activity]" are shown in Fig. 10, which are used to evaluate the text-image alignment of SRRL.

### 11.2  PHYSICAL PHENOMENON RELATED PROMPTS

Physical phenomenon related prompts are shown in Fig. 11, which are used to evaluate the image reasoning ability of SRRL.

"a cat washing dishes",
"a dog washing dishes",
"a horse washing dishes",
"a monkey washing dishes",
"a rabbit washing dishes",
"a zebra washing dishes",
"a spider washing dishes",
"a bird washing dishes",
"a sheep washing dishes",
"a deer washing dishes",
"a cow washing dishes",
"a goat washing dishes",
"a lion washing dishes",
"a tiger washing dishes",
"a bear washing dishes",
"a raccoon riding a bike",
"a fox riding a bike",
"a wolf riding a bike",
"a lizard riding a bike",
"a beetle riding a bike",
"a ant riding a bike",
"a butterfly riding a bike",
"a kangaroo playing chess"

"a fish riding a bike",
"a shark riding a bike",
"a whale riding a bike",
"a dolphin riding a bike",
"a squirrel riding a bike",
"a mouse riding a bike",
"a rat riding a bike",
"a snake riding a bike",
"a turtle playing chess",
"a frog playing chess",
"a chicken playing chess",
"a duck playing chess",
"a goose playing chess",
"a bee playing chess",
"a pig playing chess",
"a turkey playing chess",
"a fly playing chess",
"a llama playing chess",
"a camel playing chess",
"a bat playing chess",
"a gorilla playing chess",
"a hedgehog playing chess",
"a kangaroo playing chess"

Figure 10: Prompts of the template "a(n) [animal] [activity]".

"Dominoes falling to demonstrate cause and effect logic.",
"Detective connecting clues on a corkboard with string.",
"Ball rolling down ramp, showing gravity in action.",
"Magnets attracting and repelling on a metal surface.",
"Student balancing a scale with diverse weights.",
"Pendulum swinging, illustrating conservation of energy.",
"Child testing objects' buoyancy in a water tank.",
"Person decoding a ciphered message on paper.",
"Scientist comparing plant growth with and without sunlight.",
"Teacher drawing a Venn diagram to explain logic.",
"Robot sorting colored blocks by shape and hue.",
"Person solving a Sudoku puzzle on a desk.",
"Two kids racing toy cars on different surfaces.",
"Person tracing electrical circuits with a tester.",
"Detective examining fingerprints with a magnifying glass.",
"Student observing chemical reactions in test tubes.",
"Person reflecting light using mirrors onto targets.",
"Detective piecing together torn letter fragments.",
"Person solving a logic puzzle on a chalkboard.",
"Student testing friction with objects on ramps."

Figure 11: Physical phenomenon related prompts.

## 11.3 UNCONVENTIONAL PHYSICAL PHENOMENA PROMPTS

Unconventional physical phenomena prompts are shown in Fig. 12, which are used to evaluate the image reasoning ability of SRRL.

Physical phenomenon related prompts and unconventional physical phenomena prompts are provided by GPT-4o. The prompts are: "Please help me think of some prompts generated from images that

"A ball rolling uphill against gravity, surprising onlookers.",
"Dominoes falling in reverse, standing themselves up.",
"Detective finds a footprint leading to a floating shoe.",
"A magnet attracts plastic objects, not metal ones.",
"Shadow points away from the light source, defying logic.",
"A plant grows upside down, roots in the air.",
"Water flows upward in a tilted glass, defying gravity.",
"A clock runs backward, time reversing for everyone.",
"A mirror reflects a different object than in front.",
"Ice cubes float in hot coffee, not melting.",
"A candle flame burns downward, not upward.",
"Objects fall slower in a vacuum than in air.",
"A book remains dry underwater, pages untouched.",
"A person walks through a solid wall unharmed.",
"Raindrops fall upward from the ground to the sky.",
"A compass needle spins wildly, never settling north.",
"A shadow appears without any object present.",
"A person lifts a heavy rock effortlessly, surprising others.",
"A glass shatters before being touched by the ball.",
"A balloon sinks in air, going downwards rapidly."

Figure 12: Unconventional physical phenomena prompts.

demonstrate logical reasoning, in English, and output in JSON format: [prompt1, prompt2,...]. Please provide me with 20 prompts" and "Please help me think of some prompts generated from images that demonstrate reasoning and unconventional phenomena. They should be in English and output in JSON format: [prompt1, prompt2,...]. Please provide me with 20 prompts.".

## 12   LIMITATIONS

We introduce three types of reward models, CLIP Score (Hessel et al., 2021), ImageReward (Xu et al., 2023), and VQAScore (Lin et al., 2024) in the training process. However, these reward models are usually used to enhance text-image alignment or align with human feedback. Training a reward model of higher quality is of great significance for enhancing the reasoning ability of image generation models. We will reserve this for future work. Introducing better reward models can improve the accuracy of the reward function, leading to the broader application of reinforcement learning in image generation.

## 13   THE USE OF LARGE LANGUAGE MODELS (LLMS)

We use the LLM to polish the manuscript in order to enhance its readability. This includes correcting word choice and grammatical errors in the paper to achieve more polished writing.

