# OpenReview forum: "Self-Reflective Reinforcement Learning for Diffusion-based Image Reasoning Generation"
_ICLR.cc/2026/Conference — ICLR 2026 Conference Withdrawn Submission_

### Official Review · Reviewer_q5CG · 2025-10-30

**Soundness:** 3
**Presentation:** 3
**Contribution:** 2
**Rating:** 2
**Confidence:** 3

**Summary:**

The paper proposes SRRL, a self-reflective reinforcement learning algorithm for diffusion-based image generation, specifically targeting logical reasoning in images. Inspired by Chain-of-Thought (CoT) methods in LLMs, SRRL treats the entire denoising trajectory as a reasoning step, introducing multi-round reflective denoising and a condition-guided forward process. This approach allows diffusion models to generate images that adhere to physical laws and counterintuitive phenomena.

**Strengths:**

- The paper is well-written with a clear and logical structure.

- The work innovatively applies CoT to diffusion models to achieve reasoning generation.

- Quantitative results show notable improvements in metrics like CLIP Score, ImageReward, and VQAScore.

**Weaknesses:**

- A critical limitation is that all employed reward models and metrics primarily focus on semantic alignment between images and prompts, lacking the ability to capture fine-grained differences in reasoning generation of logical images. In other words, while the paper proposes a novel image reasoning generation task with interesting cases, there is a lack of direct evidence that the framework actually enables the model to learn reasoning, rather than merely generating semantically consistent images.

- The experimental section is relatively weak. It only compares against the basic DDPO baseline without benchmarking against specialized reflection-optimized models such as Reflect-DiT [1].

- The ablation studies are incomplete, e.g., the effectiveness of key components like the Condition Guided Forward Process is unvalidated.

[1] Reflect-DiT: Inference-Time Scaling for Text-to-Image Diffusion Transformers via In-Context Reflection.

**Questions:**

- Multimodal LLMs with visual understanding (e.g., GPT-4oV) or human studies may help better evaluate and improve logical reasoning in generated images.

---

### Official Review · Reviewer_dWsd · 2025-10-30

**Soundness:** 2
**Presentation:** 1
**Contribution:** 1
**Rating:** 2
**Confidence:** 4

**Summary:**

The paper introduces a method for guiding text-to-image generation with reward models, mimicking the chain-of-thought methodologies that have become widespread in the LLM field. Using pretrained diffusion models, the authors propose fine-tuning the generative model using a reward model, through multiple generation trajectories created using DDIM inversion. The authors suggest that this enables the model to reflect and improve upon previous generation trajectories. The paper includes examples of unconventional prompts and results generated by different trajectories of the proposed method, showing improving alignment with the progression of the method.

**Strengths:**

* The topic of the paper is highly valuable. Progress in aligning text-to-image models with user’s preferences has wide impacts.
* The concept of using inversion in reward-model-guided generation is novel to the best of my knowledge.

**Weaknesses:**

* The writing clarity could be improved. The main method was difficult to understand from the paper. The training objective should be stated explicitly and clearly. Including the training algorithm in the main paper (even in a more concise form) would greatly assist in the readability of the method. Also, key pieces of information included in the appendix are not referred to in the text, for example the list of prompts used to create Tab. 1.
* The authors do not include related work on test-time scaling. While test-time scaling does not change the model’s weights, the use of reward model for guided generation as well as injecting noise repeatedly has been explored in many test-time scaling works, including [1] [2] [3].
* The qualitative results are not highly convincing, and do not necessarily improve with the progression on the method (Figs. 3 and 4). Moreover the results become saturated, perhaps due to the repeated use of CFG. I believe the claim that the model’s results are on par with those generated by other advanced models is not supported by the limited qualitative evidence (personally I prefer the outputs of Nano-Banana or GPT-4o out of the options).
* The quantitative results are problematic for several reasons. First, the models were tested on a small set of prompts instead of using widely accepted benchmarks. Second, the same metrics were used both for the reward models as well as for evaluation, raising concerns of “reward hacking”. Finally, very few alternative methods are considered.

[1] Kim, Sunwoo, Minkyu Kim, and Dongmin Park. "Test-time alignment of diffusion models without reward over-optimization." arXiv preprint arXiv:2501.05803 (2025).

[2] Singhal, Raghav, et al. "A general framework for inference-time scaling and steering of diffusion models." arXiv preprint arXiv:2501.06848 (2025).

[3] He, Haoran, et al. "Scaling Image and Video Generation via Test-Time Evolutionary Search." arXiv preprint arXiv:2505.17618 (2025).

**Questions:**

* From Fig. 2, it seems that the reward model compares across samples generated with the same prompt, but in Alg. 1 it seems that different prompts are used. Could the authors clarify this point?

---

### Official Review · Reviewer_MpRA · 2025-10-31

**Soundness:** 2
**Presentation:** 2
**Contribution:** 1
**Rating:** 2
**Confidence:** 3

**Summary:**

This paper introduct reinforcement learning to diffusion models to improve reasoning capabilities. The paper proposes SRRL, which uses a "multi-round reflective denoising process" and a "condition guided forward process" to iteratively refine the generation policy across multiple trajectories. Results show SRRL improves performance on metrics like VQAScore and ImageReward compared to baselines (Vanilla SD, DDPO).

**Strengths:**

- Mapping CoT reasoning to an iterative, trajectory-wise refinement process for diffusion is a novel concept.

- Addresses the critical challenge of infusing generative models with reasoning.

**Weaknesses:**

- Uses reward models (VQAScore, ImageReward) built for alignment, not for the target task of physical reasoning. Also, same reward models are used for evaluations, prone to reward hacking.

- The multi-round, re-noising (DDIM inversion) design is extremely costly.

- The model is evaluated on the same prompts it was trained on, raising concerns about overfitting vs. true generalization.

**Questions:**

- What was the total wall-clock training time (in GPU hours) required to fine-tune a model for one experiment?

- Can you discuss core difference to Flow-GRPO [A]?

[A] Liu et al., Flow-GRPO: Training Flow Matching Models via Online RL, NeurIPS 2025.

---

### Official Review · Reviewer_Rdcf · 2025-10-31

**Soundness:** 2
**Presentation:** 2
**Contribution:** 2
**Rating:** 2
**Confidence:** 4

**Summary:**

The paper proposes Self-Reflective Reinforcement Learning (SRRL), which introduces Chain-of-Thought reasoning into diffusion models to enable self-reflective image generation. Through multi-round denoising and reinforcement learning optimization, SRRL allows models to produce images consistent with physical laws and logical reasoning.

**Strengths:**

* The paper addresses a novel problem.
* The paper is clearly written and easy to follow.

**Weaknesses:**

* Lack of analysis of computational overhead. Intuitively, even in the forward process, the model needs to perform k complete inferences and k-1 VQA model inferences, which is extremely expensive.
* Some images are difficult to understand. In Figure 5, both downward and rightward directions are labeled as *Round ↑* . This makes it difficult to understand the actual process. And the figure doesn't convey the conclusion being described.
* The specific value of K used in forward process is not clearly stated, making the improvement difficult to assess.
* Lack of detailed impact of the K value on the results. In Figure 7, the paper mentions that the quality of generated images improves as the number of reflection rounds increases, and observes the phenomenon of *Reflection Refinement*. However, the paper does not provide a detailed analysis, and the experiments are limited to only two case studies.

**Questions:**

* What specific model is used in the paper to obtain the VQAScore?
* According to my understanding, the step-by-step PPO training in this paper still relies on alignment-based metrics (VQAScore). Why can the use of the alignment metric improve the "reasoning" ability?

---

### Note · Authors · 2025-11-14

I have read and agree with the venue's withdrawal policy on behalf of myself and my co-authors.